# An Overview of the Algae-Mediated Biosynthesis of Nanoparticles and Their Biomedical Applications

**DOI:** 10.3390/biom10111498

**Published:** 2020-10-30

**Authors:** Rimsha Chaudhary, Khadija Nawaz, Amna Komal Khan, Christophe Hano, Bilal Haider Abbasi, Sumaira Anjum

**Affiliations:** 1Department of Biotechnology, Kinnaird College for Women, Lahore 54000, Pakistan; rimshachaudhary.09@gmail.com (R.C.); khadijanawaz95@gmail.com (K.N.); aaykay28@gmail.com (A.K.K.); 2Laboratoire de Biologie des Ligneux et des Grandes Cultures (LBLGC), INRAE USC1328, Université d’Orléans, 28000 Chartres, France; hano@univ-orleans.fr; 3Le Studium—Institute for Advanced Studies, 1 Rue Dupanloup, 45000 Orléans, France; bhabbasi@qau.edu.pk; 4Department of Biotechnology, Quaid-i-Azam University, Islamabad 54000, Pakistan

**Keywords:** algae, green synthesis, nanoparticles, iomedical applications

## Abstract

Algae have long been exploited commercially and industrially as food, feed, additives, cosmetics, pharmaceuticals, and fertilizer, but now the trend is shifting towards the algae-mediated green synthesis of nanoparticles (NPs). This trend is increasing day by day, as algae are a rich source of secondary metabolites, easy to cultivate, have fast growth, and are scalable. In recent era, green synthesis of NPs has gained widespread attention as a safe, simple, sustainable, cost-effective, and eco-friendly protocol. The secondary metabolites from algae reduce, cap, and stabilize the metal precursors to form metal, metal oxide, or bimetallic NPs. The NPs synthesis could either be intracellular or extracellular depending on the location of NPs synthesis and reducing agents. Among the diverse range of algae, the most widely investigated algae for the biosynthesis of NPs documented are brown, red, blue-green, micro and macro green algae. Due to the biocompatibility, safety and unique physico-chemical properties of NPs, the algal biosynthesized NPs have also been studied for their biomedical applications, which include anti-bacterial, anti-fungal, anti-cancerous, anti-fouling, bioremediation, and biosensing activities. In this review, the rationale behind the algal-mediated biosynthesis of metallic, metallic oxide, and bimetallic NPs from various algae have been reviewed. Furthermore, an insight into the mechanism of biosynthesis of NPs from algae and their biomedical applications has been reviewed critically.

## 1. Introduction

The last few decades have witnessed accelerated development in the field of nanotechnology attributed to the exceptional physio-chemical properties of nanoparticles (NPs) and associated nanomaterials. Nanotechnology has infiltrated almost every discipline of science to create novel alternatives aimed at solving different research-related bottlenecks [1]. The origin of nanobiotechnology is the fabrication of nanoscale particles by virtue of biological moieties that can influence the physical characteristics of NPs [2]. Synthesis of NPs of diverse sizes and shapes has underpinned great interest due to their novel properties as compared to their bulk counterparts. Consistency in the chemical, biochemical, and physicochemical properties of materials varies immensely at the nanoscale mainly due to the high aspect ratio of surface area to volume [3,4]. This leads to considerable differences in mechanical properties, melting point, optical absorption, thermal, electrical conductivity, biological, and catalytic activities [5]. NPs bridge the gap between bulk materials and atomic or molecular structures. NPs usually ranging in size from 1 to 100 nm and have been classified on the basis of origin, dimensions, and structural configurations, as shown in Figure 1 [6].

Due to unique morphological (shape, size, and charge distribution) and physico-chemical properties of NPs, they find applications in almost every discipline of science, like space, energy, defense, communication, biomedicine, and agriculture. Use of NPs in disease diagnosis, imaging, treatment, drug delivery, tissue engineering, and cancer therapeutics is also increasing tremendously [7]. However, application of NPs in biological entities, especially humans, is very critical, and the NPs used in biomedical applications should preferably be from green sources [8]. Taking into consideration various factors, such as biocompatibility, bioavailability, bio-distribution, and most importantly biosafety of NPs, the current trend of synthesis of NPs is moving towards the safer side [9].

NPs are mostly synthesized by physical and chemical methods such laser ablation, mechanical milling, electro explosion, and sol-gel, but they are hazardous for environmental and human health [10]. The trend to synthesize NPs from alternative, environmentally safe, cost-effective, and non-toxic green methods is increasing [11]. Organisms such as algae, fungi, bacteria, and plants allow green synthesis of NPs that are free from impurities [12,13]. These biological systems possess various biological entities which behave as both reducing and capping agents in NP generation, either intracellularly or extracellularly [14]. Proteins, peptides, and functional groups (carbonyl, thiol, hydroxyl, and amine) present in biological entities reduce precursor salt into NPs called reducing or stabilizing agents [15]. They cap and stabilize NPs at the same time, hence this type of synthesis is relatively simple, reproducible, and stable [9].

The biosynthesis of NPs can be controlled by various physical factors, such as pH, concentration of metal precursor and bio-extract, contact time of reaction mixture, exposure to any light source, and temperature. These factors control the nucleation, growth, aging, agglomeration, and stabilization of NPs, which in turn can alter the size, shape, and crystallinity of the synthesized NPs [16,17,18,19]. After synthesis of NPs, they are subjected to various microscopic and spectroscopic characterization techniques to find out their unique physico-chemical characteristics. Determination of the exact morphological (size, shape, and particle distribution) and physico-chemical properties of NPs is very critical in their biomedical applications [20]. The most commonly employed microscopic techniques used to find out the size and morphological properties of NPs include transmission electron microscopy (TEM), scanning electron microscopy (SEM), atomic force microscopy (AFM), and high-resolution transmission electron microscopy (HR-TEM), and are employed to determine the size and morphological features of NPs [21,22]; whereas to analyze the structure, composition, and crystallinity of synthesized NPs, UV-visible spectroscopy (UV-Vis), energy dispersive spectroscopy (EDS), selected area electron diffraction (SAED), dynamic lights scattering (DLS), X-ray diffraction (XRD), X-ray photo-electron spectroscopy (XPS), and fourier transform infrared spectroscopy (FTIR) are used [23,24].

For green synthesis, the shift towards the use of algae is increasing, as they are a rich source of secondary metabolites, proteins, peptides, and pigments, which serve as nano-biofactories [25]. Moreover, they have fast growth rate, easy harvesting, and cost-effective scale-up, making them good candidates for biological synthesis of NPs [26]. Algae are the most primitive organisms inhabiting diverse ecosystems and dominant photosynthetic organisms on Earth [27]. Algae possess the ability to hyper-accumulate metals and transform them into NPs, making them the best choice for green synthesis [9]. Many metallic and metal oxides NPs are synthesized from various classes of algae, such as blue-green algae (Cyanophyceae), brown algae (Phaeophyceae), green algae (Chlorophyceae), and red algae (Rhodophyceae), as shown in Figure 2 [28,29].

Algae have long been exploited commercially and industrially as food, feed, additives, cosmetics, pharmaceuticals, fertilizer, and bioremediation agents. The rationale behind the use of algae for green synthesis is that no external reducing or capping agents are required, low energy input, low cost synthesis, and simple reaction. However, algal use for NPs synthesis, a branch known as phyco-nanotechnology, is still in its infancy. Henceforth, this review critically reviews the potential of algae-based biosynthesis of NPs, mechanisms involved in their synthesis, and significant industrial/biomedical applications of these algae-mediated NPs.

## 2. Mechanism Involved in the Algae-Mediated Biosynthesis of NPs

Algae are known to hyper-accumulate heavy metal ions and possess an exceptional capability to remodel them into more malleable forms. Because of these alluring attributes, algae have been foreseen as model organisms for fabricating various types of nanomaterials, especially metallic NPs [27]. The biosynthesis of NPs is initiated when a precursor metal solution is incubated with algal extract. The biochemical compounds in algae, such as antioxidants, pigments, phycobilins, chlorophylls, oils, minerals, carbohydrates, fats, proteins, polyunsaturated fatty acids, and various phytochemicals, aid in reducing charge of metal ion to zero valent state. Bio-reduction is a three-phase process, in which the activation phase involves metal ion reduction and nucleation due to the enzymes secreted by algal cells evident from the color change of solution. In the growth phase, nucleated metal elements amalgamate with each other, forming NPs of different sizes and shapes which are thermodynamically stable. The final shape of NPs is acquired in final termination phase. Factors such as temperature, pH, time, static conditions, substrate concentration, and stirring control the physical characteristics of NPs [30]. Herein, we have specifically discussed the mechanism involved in algae-mediated biosynthesis of AuNPs, which is almost same as that involved in synthesis of other NPs.

Biosynthesis of NPs from algae could either be intracellular or extracellular depending on the location of NPs formed. The intracellular method is a dose-dependent process in which NP biosynthesis occurs inside the algal cell, as shown in Figure 3. NADPH or NADPH-dependent reductase released during the metabolic processes such as nitrogen fixation, photosynthesis, and respiration act as reducing agents [31,32]. AuNPs were synthesized intracellularly by incubating chloroauric acid with *Ulva intestinalis* and *Rhizoclonium fontinale* algae at 20 °C for 72 h. A visible purple color change of thallus from green indicated the biosynthesis of AuNPs. Furthermore, incubating gold metal solution with biomass resulted in no color change, which established that no intracellular enzyme or metabolites were associated with the bio-reduction process. In another experiment, the silica gel suspension encapsulated *Klebsormidium flaccidum* showed a visual purple color change of chloroplast from green, which is an indication of the potential of cells to reduce gold precursor (chloroauric acid). This was further confirmed by TEM analysis, which showed dark-colored spots within the thylakoid membrane, indicating the presence of reduced gold precursor salt by the NADPH-dependent reductase enzyme or NADPH [33]. Similarly, Senapati and co-workers (2012) demonstrated the intracellular synthesis of AuNPs by the algal cell wall in *Tetraselmis kochinensis*. UV-visible spectroscopy clearly proved that there was no extracellular synthesis. The AuNPs were more densely present near the cell wall rather than the cytoplasmic area, which is most likely due to the presence of bioactive moieties responsible for bioreduction [34].

The extracellular mode of synthesis occurs when metal ions become attached to the surface of algal cells and metabolites such as proteins, lipids, non-protein RNA, DNA, ions, pigments, and enzymes reduce them at the surface [35]. The extracellular mode of synthesis is more convenient as NPs are easily purified, however some essential pre-treatments like washing and blending of algal biomass are required [36]. Certain physio-chemical conditions like pH, temperature, and type and initial concentration of metal and substrate influence the size, shape, and agglomeration of NPs [32,37]. Higher pH prevents the agglomeration of NPs by enhancing the reducing power of functional groups [18,38]. Basic pH helps in the capping and stabilization of NPs by interacting with the amine groups of surface-bound proteins and their residual amino acids [39]. Extracellular synthesis of AuNPs synthesized by *S. platensis* at varying concentrations of chloroauric acid was confirmed by the presence of surface plasmon peak at 530 nm, which gives a clue of the involvement of proteins, enzymes, and biomolecules in algae-mediated synthesis of NPs [40].

## 3. Biosynthesis of NPs from Algae

Algae are a diverse group of photoautotrophic, eukaryotic, aquatic, unicellular/multicellular organisms and have been classified on the basis of the pigmentation they release, which includes brown algae (phaeophytes), red algae (rhodophytes), and green algae (chlorophytes) [41,42]. Algae are commonly used for the biosynthesis of various metallic and metal oxide NPs, because they grow rapidly, are easy to handle, and their biomass growth, on average, is ten-times faster than higher plants. Different algal strains have been researched for the green synthesis of different types of NPs to date. Herein, the literature is critically reviewed for biosynthesis of NPs from different classes of algae in detail.

### 3.1. Brown Algae-Mediated Biosynthesis of NPs

Brown algae belongs to the order Fucales and family Sargassaceae. The dominant components of Fucales are sterols such as cholesterols, fucosterols, sulfated polysaccharides, and functional groups like glucuroronic acid, muramic acid, alginic acid, and vinyl derivatives which act as reducing as well as capping agents for the synthesis of NPs [43]. Currently, various metallic (silver and gold) and metal oxide (Zinc oxide and titanium oxide) NPs have been synthesized from different species of brown algae, as discussed in Table 1.

Metallic NPs such as silver (AgNPs), gold (AuNPs), and copper (CuNPs) are some of the most widely synthesized NPs from brown algae [11,44,45,50,51]. Among different metallic NPs, more than half of the reported data in literature are about the synthesis of AgNPs from different algae strains. This is because the AgNPs possess superior physico-chemical characteristics as compared to their bulk forms, thus making them extremely useful in different industries, such as in jewelry, paints, textile, dental alloys, drug-delivery, and wound-healing [76,77,78]. In order to biosynthesize AgNPs from brown algae, a variety of species have been reported in literature, such as *Gelidiella acerosa*, *Turbinaria conoides*, *Desmarestia menziesii*, *Sargassum polycystum*, *Padina pavonica*, and *Cystophora moniliformis* [11,44,45,50,51]. In one report, spherical AgNPs (96 nm) have been synthesized extracellularly from *T. conoides*, which exhibited tremendous anti-bacterial activity against *Staphylococcus aureus*, *Staphylococcus epidermis*, *Escherichia coli*, *Candida albicans*, *Aspergillus niger*, and *Pseudomonas aeruginosa* [44]. The organic moieties, amines, polyamines, free hydroxyl, and carbonyl groups of *Turbinaria* species (*T. ornate* and *T. conoides*) have been reported to act as reducing agents of precursor silver salts used in the synthesis of AgNPs [48,79].

The other type of NP that has been extensively synthesized from brown algae strains is AuNPs, which has exhibited a number of medicinally important bio-activities, such as anti-fouling, anti-coagulant, and anti-bacterial activities [47,50,80]. Among all reported species of brown algae, *T. conoides* is one of the most prominent types that are traditionally used in the generation of AuNPs. A variety of shapes, like polydispersed, rectangular, spherical, and triangular AuNPs, were generated from *T. conoides* by extracellular pathway [81]. In most of the cases, AuNPs were synthesized by using chloroauric acid as a precursor of gold ions along with *T. conoides*. Another important species of brown algae, *Laminaria japonica*, has also been investigated in the green synthesis of AuNPs. *L. japonica* is a rich source of bio-active components such as polyphenols, peptides, proteins, vitamins, carotenoids, and fibers [80]. Spherical (15–20 nm) AuNPs were extracellularly synthesized from *L*. *japonica* with the involvement of phytochemicals and functional groups that behaved as reducing and capping agents [50]. Other species of brown algae, such as *Fucus vesiculosus*, *Sargassum myriocystum*, *Ecklonia cava*, *Sargassum wightii*, *Stereospermum marginatum*, *Padina gymnospora*, *Dictyota bartayresianna*, and *Cystoseira baccata*, have also been reported in the biosynthesis of AuNPs (Table 1).

In addition to the metallic NPs, brown algae have also been reported for biosynthesizing various metal oxide NPs, such as zinc oxide nanoparticles (ZnONPs) and titanium oxide nanoparticles (TiO_2_NPs) [82]. According to one study, ZnONPs were synthesized by mixing dried algal powder from *S. muticum* with distilled water and heated until completely mixed, then zinc acetate salt solution was added, and it was placed on continuous stirring for hours until the generation of NPs. The synthesized ZnONPs were hexagonal in shape, ranging from 35 to 57 nm in size, and were capped by bioactive functional groups like sulfate, amines, hydroxyl, and carbonyl [11,83].

### 3.2. Red Algae-Mediated Biosynthesis of NPs

Red algae belong to the family *Rhodophyta* and are primarily used as food in many countries due to their unique flavour and the richness of several important vitamins and proteins [84]. These vitamins and proteins could be the best contenders for reduction and stabilization in algae-mediated biosynthesis of NPs. However, the synthesis of NPs from seaweed red algae is still in the developmental stages due to of its self-aggregation, slow crystallization growth, and stability issues [54,85]. Among various red algae strains, *Porphyra vietnamensis* is one of the most evident species which has been reported numerous times for synthesis of various types of NPs, due to the presence of a strong reducing agent, such as sulfated polysaccharides, that contain anionic disaccharides units, comprised of 3-linked-d-galactosyl residues flashing with 4-linked 3,6-anhydro-l-galactose and 6-sulfate residues [86,87]. A number of red algae strains have been reported in literature for the biosynthesis of AgNPs, such as *Kappaphycus alvarezii*, *Palmaria decipiens*, *Gelidiella acerosa*, *Gracilaria dura*, *Kappaphycus* sp., and many others, as summarized in Table 2.

Red algae-mediated AgNPs are efficient, eco-friendly, less time consuming, and cost-effective in comparison to the physio-chemical method [108]. The size and shape of NPs are the crucial factors that matter a lot in biomedical applications. It has been reported that the AgNPs synthesized from different red algae strains are mainly spherical in shape and ranging in size from 20 to 60 nm [88]. These extracellularly synthesized AgNPs show anti-microfouling activity, which is of great interest in the medical field [88,109]. It requires a number of steps in the intracellular synthesis of NPs to evacuate microfouling, but the prevention of microfouling in extracellular synthesis is a single step process [110]. *Gelidium amansii* is another red algae that is reported in the biosynthesis of AgNPs as well as in the minimization of micro-fouling by the 96-well method [89,111].

A limited number of studies have been carried on the red algae-mediated biosynthesis of AuNPs in comparison to AgNPs. *Lemanea fluviatilis* is one such marine red alga that is investigated for the biosynthesis of AuNPs, using chloroauric acid as a precursor salt. It generated face-centered cubic and poly-dispersed crystalline AuNPs of 5.9 nm in size which were observed by TEM [112,113]. In another report, *Corallina officinalis* has also been reported for extracellular synthesis of spherical AuNPs with reducing agents like hydroxyl, phenol, and carbonyl functional groups [95]. In addition to *L. flaviatilis* and *C. officinalis*, many other red algae species, such as *Kappaphycus alvarezii*, *Galaxaura elongata*, and *Chondrus crispus*, have been reported for facile biosynthesis of AuNPs [92]. Besides monometallic NPs, the red algae strain *Gracilaria edulis* efficiently biosynthesized bimetallic Ag-Au NPs by using different molar ratios (1:1, 1:3, and 3:1) of AgNO_3_ and HAuCl_4_ [107]. These synthesized bimetallic NPs have exhibited potent anticancerous activities against human breast cancer lines.

### 3.3. Blue-Green Algae-Mediated Biosynthesis of NPs

Blue-green algae have an anomalous state in the biological world and belong to the order of *Chroococcales*, which has two distinct families of *Chroococcaceae* and *Entophysalidaceae*. The members of these two families are distinguished by their growth habitat forming colonies [114]. They grow in dense patterns as parenchymatous cell masses found on moist rocks. Blue-green algae are photoautotrophic in nature, as they use water as an electron donor and contain two photo-pigments, chlorophyll *a* and carotene, which help in photosynthesis. On the basis of their morphology, they are also considered counterparts of unicellular bacteria [115]. Unlike brown and red algae, blue-green algae have also been widely exploited for the synthesis of various types of NPs, as shown in Table 3.

The major contributor of AgNPs by blue-green algae is *Spirulina platensis. S. platensis* is free floating, filamentous cyanobacteria that has multicellular trichomes with one open end and left-handed helix [135]. It also contains 60–70% vegetable protein, which is rich in essential amino acids, beta carotene, iron, natural vitamins, and essential fatty acids, which can help in reduction and capping of NPs [128]. Spherical-shaped AgNPs (2–8 nm) have been synthesized from *S. platensis,* which are efficiently used in pharmaceuticals, health, and food industries. Besides *S. platensis*, AgNPs of different sizes and shapes have also been synthesized from various other blue-green algae species, such as *Oscillato riawillei* (spherical, 10–25 nm)*, Plectonema boryanum* (octahedral, 200 nm), *Microchaete diplosiphon* (spherical, 80 nm), and *Cylindrospermum stagnale* (pentagonal, 38–88 nm) [132].

Like AgNPs, *S. platensis* has also shown its contribution in the biosynthesis of AuNPs. Various researchers have reported the *S. platensis*-mediated extracellular synthesis of spherical, octahedral, and cubic AuNPs, showing the involvement of proteins and peptides as reducing agents [40,136]. *Phormidium valderianum* is another important type of blue-green algae, known as alkalo-tolerant *Rhodococcus*, which has generated intracellular mono-dispersive triangular AuNPs at wavelength of 530 nm with absorbance 1.897 by UV-Vis spectrometry [124]. The appearance of one broad peak of AuNP at 530 nm is because of surface plasmon resonance (SPR), which depends on particle size, constant di-electric medium, and surface absorbed species [137,138]. *P. valderianum* has also been reported for extracellular synthesis of spherical, hexagonal, and FCC (24 nm) AuNPs by using cytoplasmic metabolites as reducing agents [38,115]. Besides monometallic NPs, *S. platensis* has also been reported in the biosynthesis of bimetallic NPs, such as core shell Ag-AuNPs and magnetic crystalline-shaped silica-NPs with the help of extracellular proteins [62,133]. *Chlamydomonas reinhardtii*, another important fresh-water green algae species, was reported to be involved in the mediation of cadmium sulfide bimetallic nanoparticles (CdSNPs) [134]. CdS belongs to the II-VI group of semiconductors, possesses unique optoelectronic properties, and has been widely used in photo-catalysis, LEDs, and biosensors.

### 3.4. Green Algae-Mediated Biosynthesis of NPs

Green algae are divided into two main groups on the basis of their habitat as micro and macro green algae [139]. Micro green algae are unicellular and mainly cultivated/habitat in fresh water, whereas macro green algae are multicellular marine-living plant-like organisms [140]. Biosynthesis of various monometallic, bimetallic, and metal oxide NPs from green algae is currently widely practiced [58]. Herein, we have separately described the biosynthesis of NPs from both types of green algae as micro-mediated and macro-mediated biosynthesis.

#### 3.4.1. Green Micro Algae-Mediated Biosynthesis of NPs

Micro green algae belong to the order *Cladophorales* and have been extensively used in various industrial, health, and biotechnological applications. They are a rich source of many essential components, such as alkaloids, phenols, flavonoids, carbohydrates, and functional groups that could behave as reducing as well as stabilizing agents in micro-mediated biosynthesis of NPs [141]. Among various monometallic NPs, AgNPs are most extensively in-vitro generated NPs from different species of micro green algae, as shown in Table 4.

To date, more than 20 different species of green micro algae have been exploited for biosynthesis of AgNPs. The AgNPs synthesized from different species exhibit interesting and variable physico-chemical characteristics when analyzed by different spectroscopic and microscopic techniques, such as SEM, XRD, FTIR, DLS, and EDX [13,47,143,144,145]. Almost all micro green algae species used extracellularly generate AgNPs of varying size and morphology, such as *Pithophora oedogonia*, (cubical and hexagonal, 24–55 nm), *Chlorococcum humicola* (spherical, 16 nm), *Chlorella vulgaris* (triangular, 28 nm), *C. reinhardtii* (rectangular and rounded, 1–15 nm), and *Enteromorpha flexuosa* (circular, 15 nm) [141,145,149,151]. Similar to AgNPs, a lot of data have been published in recent years on green micro algae-mediated biosynthesis of AuNPs, as mentioned in Table 4. *Pithophora crispa* from higher altitude is one of the most widely exploited species of micro algae involved in the biosynthesis of AuNPs by reducing chloroauric acid precursor salt with the help of intracellular and extracellular proteins and peptides [146]. The majority of the reported primarily metabolites involved in the biosynthesis of metallic NPs from green micro algae are proteins, peptides, cyclic compounds, and carboxylic acids [92,158,161].

Apart from AgNPs and AuNPs, micro green algae have also been used for the synthesis of semiconductor NPs. Many attempts have been done to synthesize silicon-NPs from micro green algae. Silicon-NPs are semiconductor in nature and are used as bio-indicators in many industrial wastes to identify the presence of toxic compounds. They also possess an important position in the ecological cycle, as they play essential roles in oxygen production and nutrient recycling. For this, green algae *C. vulgaris* was taken into account, silicon alkaloids were used as silicon precursor, and were mixed with algal extract. Silicon-NPs were synthesized by hydrolysis and poly-condensation of silicon alkaloids by peptides and proteins present in *C. vulgaris* extract [142]. Other than silicon-NPs, a number of other metallic, bi-metallic, metal oxide, and semiconductor NPs’ biosynthesis is in process and a lot of research and experiments are in their exponential phases.

#### 3.4.2. Green Macro Algae-Mediated Biosynthesis of NPs

Green macro algae are also called bio-factories for the synthesis of metallic NPs, because they possess numerous valuable compounds which are responsible for reduction and capping of NPs [58,163]. Besides being involved in the synthesis of NPs, these compounds also exhibit strong anti-tumor, anti-viral, anti-bacterial, and cytotoxic effects to microbes [164]. In recent trends, various green macro algae strains have been extensively used in the generation of metallic NPs, as summarized in Table 5.

*Ulva fasciata* is one of most useful green macro algae species, and was utilized to generate nano-sized silver colloids which were further applied to cotton fabric in the presence and absence of citric acid in order to evaluate their anti-microbial efficacy [68]. In another report, *Gracilaria edulis* (rich in amide, carboxylic, and nitro compounds) was used for the synthesis of spherical AgNPs and octahedral ZnONPs [163,168]. *Chaetomorpha linum*, another important species of seaweed green macro algae, is well recognized for its ecological roles in the regulation of nutrients availability to its habitat, has been also used for the synthesize of AgNPs by catalyzing the reduction of silver ions (Ag^+^) to Ag^0^ with the help of peptides, flavonoids, and terpenoids in extracellular environment [163]. Besides AgNPs, AuNPs have also been synthesized by green macro algae species such as *Prasiola crispa* and *Rhizoclonium fontinale* [65,125]. During the past few years AuNPs have been of great interest due to their use in targeted drug delivery in cancer treatment. There is always a challenge in AuNP synthesis because of their non-reproducibility at the perfect size and shape, but green macro algae took over this challenge and synthesized stable and reproducible AuNPs [65,125,163].

## 4. Biomedical Applications of Algae-Mediated NPs

NPs synthesized from various green approaches are usually biocompatible and free from toxic chemicals entangled on their surfaces, because they do not use any external capping or reducing agents during the synthesis of NPs and therefore show less toxicity than chemically synthesized NPs. Algae species also do not involve the use of any toxic chemicals during reduction and stabilization of NPs, as they contain naturally occurring biomolecules which impart less or no toxicity, and thus can be preferably used in various biomedical applications [165,169]. Various applications of algae-mediated NPs, predominantly in biomedicine are discussed herein in detail.

### 4.1. Antibacterial Activity

The widespread use of antibiotics to treat bacterial infections has led to the emergence of multi-drug-resistant bacterial strains. Providing a safe and efficient treatment for drug-resistant bacterial strains is a major health challenge faced globally. Therefore, there has been a shift towards the use of NPs as an alternative antibacterial agent, which has demonstrated efficient and superior bactericidal activity. Since NPs kill bacteria by disrupting the cell membrane and generating reactive oxygen species (ROS), they have broad-spectrum antibacterial activity against gram-positive and gram-negative bacteria [170].

The NPs synthesized from algae have been investigated for their antibacterial activity against a range of bacterial strains. The AgNPs biosynthesized from brown seaweed *Padina tetrastromatica* efficiently retarded the growth of *P. aeuroginosa*, *Klebsiella planticola*, *Bacillus subtilis*, and other *Bacillus* sp. [171]. In another study, stable and colloidal-shaped AgNPs prepared from the aqueous extract of green marine algae *Caulerpa serrulata* exhibited exceptional antimicrobial ability at lower concentrations against *Shigella* sp., *S. aureus*, *E. coli*, *P. aeruginosa*, and *Salmonella typhi.* The highest zone of inhibition of 21 mm of AgNPs (75 µl) was recorded against *E. coli*, whereas the smallest zone of inhibition of 10 mm at 50 μL AgNPs was observed against *S. typhi* [19]. Similarly, AgNPs synthesized from *Pithophora oedogonia* aqueous extract have shown potential antibacterial activity against *E. coli*, *Micrococcus luteus*, *S. aureus*, *B. subtilis*, *Vibrio cholerae*, *P. aeruginosa*, and *Shigella flexneri.* The highest zone of inhibition (17.2 mm) was measured for *P. aeruginosa*, which shows exceptional antibacterial activity of AgNPs against more resistant gram-negative rods [13].

Furthermore, spherical AuNPs synthesized from the protein extract of blue-green alga *S. platensis* significantly inhibited the growth of *S. aureus* and *B. subtilis* [118]. AuNPs synthesized by from *Ecklonia cava* and *Nitzschia* have been tested for their antibacterial activity against *E. coli*, *S. aureus*, *P. aeruginosa*, *B. subtilis*, *Aspergillus fumigatus*, *C. albicans*, and *A. niger* [52,172]. The AuNPs synthesized from *Stoechospermum marginatum* displayed superior antibacterial activity against *Enterobacter faecalis* as compared to tetracycline antibiotic standard [57]. In another study, *Neodesmus pupukensis*-mediated AgNPs and AuNPs were tested for their antibacterial potential against various strains of bacteria. The results showed that the zones of inhibition of AgNPs were: *Pseudomonas* sp (43 mm); *E. coli* (24.5 mm); *K. pneumoniae* (27 mm); *S. marcescens* (39 mm), while AuNPs showed activity to only *Pseudomonas* sp. (27.5 mm) and *S. marcescens* (28.5 mm) [173]. These findings show the promising application of algae-mediated NPs as antibacterial agents in future.

### 4.2. Antifungal Activity

Fungal infections are becoming a growing public health concern due to the limited availability of antifungal drugs and emerging resistance to antifungal drugs. There is a strong incentive to develop new, strong, and effective antifungal agents. NPs could be a novel treatment option for fungal infections, as they demonstrate excellent fungicidal activity [174]. AgNPs have been the most effective antifungal agent synthesized by the green approach so far. A study reported the synthesis of AgNPs from *Sargassum longifolium* which have been examined for their antifungal activities at various concentrations against different pathogenic fungal strains, including *A. fumigatus*, *Fusarium* sp., and *C. albicans.* The results showed that AgNPs significantly inhibited the growth of each fungal strain in a dose-dependent manner [175]. In another study, AgNPs were synthesized using red seaweed *Gelidiella acerosa* aqueous extract and tested for antifungal activity against *Fusarium dimerum*, *Mucor indicus*, *Humicola insolens*, and *Trichoderma reesei.* The results indicated considerable antifungal activity of AgNPs as compared to standard antifungal drug [96]. AgNPs biosynthesized from green algae *Ulva latica* and red algae *Hypnea musciformis* have been effective in retarding the growth of *A. niger*, *C. albicans*, and *Candida parapsilosis* fungal strains [176].

Algae-synthesized AuNPs have also been investigated for their antifungal activity, however only a few studies have been reported in this regard. AuNPs synthesized by using aqueous extract of brown seaweed *Dictyota bartayresiana* showed antifungal activity against soft rot fungus *F.dimerum* and *Humicola insolens* [177]. Similarly, *Neodesmus pupukensis*-mediated AgNPs and AuNPs were tested for their antifungal potential. The antifungal potency of AgNPs was confirmed with mycelial inhibition of 80.6%, 57.1%, 79.4%, 65.4%, and 69.8% against *A. niger*, *A. fumigatus*, *A. flavus*, *F. solani*, and *C. albicans*, respectively, while AuNPs had 79.4%, 44.3%, 75.4%, 54.9%, and 66.4% against *A. niger*, *A. fumigatus*, *A. flavus*, *F. solani*, and *C. albicans*, respectively [173].

### 4.3. Antifouling Agent and Biofilm Prevention

Most bacteria exist as biofilms which contain diverse species, such as fungi and algae, that interact with each other and their environment [178]. This undesirable growth on submerged surfaces is regarded as biofouling, which poses significant health risks and financial losses in marine, medical, and industrial fields [179]. Antifouling methods range from biocides to the use of toxic chemicals, however they end up accumulating and polluting the environment. Therefore, NPs have been investigated as alternative antifouling agents, as they can effectively inhibit bacterial adhesion through NP-ligand interaction and biofilm formation on surfaces, as shown in Figure 4 [180]. AgNPs have been reported to significantly prevent biofilm formation against gram-positive and gram-negative bacteria, including *E. coli*, *Salmonella* sp., *Aeromonas hydrophila*, and *S. liquefaciens.* Circular AgNPs (2–17 nm) proved lethal to *A. salina* brine shrimp, having an LC50 of 88.94 μLmL^−1^ [35]. Likewise, AgNPs synthesized from *S. ilicifolium* (33–40 nm) exhibited cytotoxicity against *A. salina* [181]. In another study, phytagel and apcomin zinc chrome paint coated with *T. ornate*-synthesized AgNPs inhibited the growth of macroflora as well as microflora. The AgNPs limited biofilm formation, with over 71.9% inhibition in *E. coli* and 40% inhibition in *Micrococcus specie*. AgNPs can also serve as antifouling agents selective towards target species; e.g., AgNPs showed a 100% mortality rate for *Balanu samphitrite* larvae hatchlings, while 56.6% for *A. marina* [74].

In addition to AgNPs, CuNPs have also been used as anti-biofilm agents against some clinical *P. aeruginosa* isolates; results showed that CuNPs not only prevented biofilm formation but also diminished hydrophobicity of cell surfaces and extracellular polymeric substances of *P. aeruginosa* [180]. In a recent study, *S. myriocystum*-mediated AgNPs of different concentrations (10, 20, 30, 40, and 50 µg/mL) were tested against biofilm-producing bacterial strains *S. epidermidis* and *P. aeruginosa*; the maximum percent of biofilm inhibition (67.75%) was obtained at 50 µg/mL conc., whereas 48.34% was obtained in 50 µg/mL AgNPs treated with *S. epidermidis*. Besides *P. aureginosa*, the inhibition of biofilm formation rate recorded was 55.49% for AgNPs at higher concentration (50 µg/mL) [182]. These findings suggest that the algal-mediated NPs can be replaceable formulations of antifouling agents in the near future and biofilm inhibition of NPs at minimum inhibitory concentrations was linked to their inhibitory effect on gene expressions connected to motility and biofilm formation [183].

### 4.4. Anti-Cancerous Activity

One of the most active areas of nano-biotechnology research includes the use of NPs for cancer therapy and for targeted delivery of anti-cancerous drugs [184]. Various studies have been published in the recent era on anti-cancerous activities of algae-mediated NPs. In a study, *Sargassum vulgare*-synthesized AgNPs (10 nm) exhibited significant anti-cancerous activity against HeLa cells and human myeloblastic leukemic cells HL60 [180]. Silver nano-triangles coated with algal-derived chitosan polymers (Chit-AgNPs) served as photothermal agents against non-small human lung cancer cell line (NCI-H460) [185]. Moreover, *S. muticum*-mediated AgNPs have shown in-vitro cytotoxic effects against MCF7 breast cancer cell line. Varying concentrations of AgNPs from 3 µg/mL to 50 µg/mL were treated with MCF7 cell line for about 48 h and the highest viability rate of 100.36% was observed at 12.5 µg/mL concentration. These AgNPs have induced ROS intracellularly, which results in the apoptosis and eventually death of cancer cells [186]. In another study, *S. myriocystum*-synthesized AgNPs were assessed for their cytotoxic abilities against the HeLa cell line by being used in various concentrations ranging from 0, 2, 4, 8, 16, 32, 64, 128, 256, to 512 µg/mL via MTT assay. Results showed that the AgNP-treated HeLa cell line showed 50% inhibitory and apoptotic activities and overall cytotoxic abilities increased with an increase in concentration of AgNPs in the medium [182]. Similarly, in vitro cytotoxicity potential of algal-mediated AgNPs against breast cancer MCF-7 cell line was observed by using various concentrations (0–100 µg/mL) for time intervals of 24, 36, and 48 h. The maximum inhibitory concentration value of AgNPs was recorded 20 µg/mL against breast cancer cells which have shown nuclear fragmentation, apoptosis, and cell death, confirming the anticancerous potential of AgNPs [187].

Algae-mediated AuNPs have also been reported to show strong anti-cancerous activities against various cell lines. In one study, *Acanthophora spicifera*-mediated AuNPs exhibited strong anti-cancerous activities against the colorectal adenocarcinoma HT-29 cell line. AuNPs were added in concentrations of 1.88, 3.75, 7.5, 15, and 30 µg/mL and incubated for about 24 h and observed via MTT assay. The maximum inhibitory concentration of 21.86 µg/mL was reported, which led to apoptosis, lack of morphological structure, and cell shrinkage in cancer cell lines [188]. Another study reported marine algae *Chaetomorpha linum*-mediated AuNPs, which showed in-vitro anti-cancerous potential against HCT-116 colon cancer cell line. Dose-dependent cytotoxic effects of AuNPs were reported in colon cancer cell lines after incubation with these nanoparticles. A series of apoptotic inductions were observed which triggered the activation of apoptotic caspase 3 and 9 along with reduction in anti-apoptotic proteins like Bcl-xl and Bcl-2, which veritably confirmed that the algal-synthesized AuNPs are efficient anti-cancerous agents [189]. Similarly, AuNPs coated with polyethylene glycol can kill maximum tumor cells as compared to an anticancer cytokine tumor necrosis factor-alpha [8,190]. These studies depict the anti-cancerous potential of algal-derived NPs.

### 4.5. Nano-Bioremediation

The use of NPs as an innovative method to remediate contaminated sites has received great attention recently. Algal-synthesized NPs have been tested as bioremediation agents; for example, *U. lactuca*-synthesized AgNPs photo-catalyzed the breakdown of methyl orange dye when illuminated with visible light. Further, it was observed that a low dosage of *U. lactuca-*mediated AgNPs greatly reduced the population of *Plasmodium falciparum*, which is otherwise a chloroquine-resistant species [89]. In a comparative study, the AgNPs synthesized from *Microchaete* showed great de-colorization ability against methyl red azo dyes as compared to cyanobacterial extract [131]. Another study showed the catalytic activity of AuNPs synthesized from the aqueous extract of brown algae *S. tenerrimum* and *T. conoides* against the organic dyes sulforhodamine, rhodamine B, and aromatic nitro compounds [60]. Similarly, the *S. myriocystum*-mediated AgNPs exhibited potential photocatalytic activity against methylene blue. The maximum percentage of MB degradation was observed at 98% within 60 min [182]. In a recent study, the *Chlorella ellipsoidea*-mediated biomatrix-loaded AgNPs exhibited high photocatalytic activity for the degradation of the hazardous pollutant dyes methylene blue and methyl orange. The catalytic efficiency was sustained even after three reduction cycles [191]. In one study, the green alga *Scenedesmus obliquus* was used to synthesize lipid-cadmium sulfide NPs. During synthesis, it was observed that the adsorption kinetics of Cd^2+^ ions was significantly increased and chemisorbed monolayer (Cd^2+^) irreversibly attached to the biomass of alga. Thus, *Scenedesmus* proved to be efficient model alga for the bioremediation of Cd^2+^ ions due to its high retention capability [192]. In this way, various algae-mediated mediated NPs are playing their positive role in remediating various kinds of heavy metals, organic/aromatic compounds, and numerous types of dyes.

### 4.6. Biosensing

Algal-synthesized AuNPs have shown great optical properties which can be utilized in biosensing applications, such as sensing the type and level of hormones in human body, particularly useful in cancer diagnostics. The algal-synthesized nano Au-Ag alloy demonstrated significant electro-catalytic ability against 2-butanone under room temperature, acting as a platform for developing an early-stage cancer detecting biosensor which can easily sense the presence of malignant cells at very initial stages [193]. In a recent report, *Noctiluca scintillans*-mediated AgNPs were evaluated for colorimetric sensing of hydrogen peroxide, which is used as an antiseptic and is indicated for small dermal scratches; in mouth, gum, teeth pain, or whitening and oral discharge. Results showed that the decomposition of hydrogen peroxide on AgNPs’ catalytic surface was found to be pH-, temperature-, and time-dependent. The test showed also a color change from brown to colorless, with hydrogen peroxide presenting the most noticeable change in color [194]. In another study, The biosynthesized AuNPs by *Hypnea valencia* also showed the ability to detect human chorionic gonadotrophin (HCG) hormone in urine samples of pregnant women during HCG blood pregnancy test [195]. Furthermore, Platinum NPs (PtNPs) synthesized from *S. myriocystum* serve as biosensors for detecting the adrenaline level in the body, which is a hormone-based drug used to treat heart attacks, asthma, and allergies [196].

## 5. Limitations and Future Prospects

There is no doubt that algae serve as excellent candidates for the green synthesis of NPs, as they are rich sources of secondary metabolites, which act as reducing and capping agents. However, this field is still in its infancy and could not be scaled up for commercial uses. This could be due to several limitations of algal-mediated biosynthesis of NPs, such as slow kinetics of reactions (taking a few days to weeks), low yield of NPs, poor morphological characteristic of biosynthesized NPs, choice of algal strain, and lack of optimization of synthesis conditions, such as pH, temperature, contact time, and concentration. Besides these, the yield of NPs is also variable, and the process control is an issue to be addressed. Moreover, colloidal stability is also often an issue that needs serious attention, because in certain cases a high level of agglomeration has been reported. A little knowledge regarding synthesis mechanism has also limited the use of alga in NP production. Therefore, to build large-scale photo-bioreactors, further research is needed to address the issues of kinetics, yield, and cell viability, along with a comparative study on the physiochemical properties of NPs synthesized by conventional methods and using algae is also a big knowledge gap that should be filled by the scientific community. Moreover, a significant amount of research is needed to identify and establish the role of specific biomolecules responsible for the reduction and capping of NPs during the algae-mediated biosynthesis process. In addition to this, only a limited number and types of NPs have been synthesized from algae, there are still other NPs like zinc oxide, palladium, silicon, and carbon-based NPs whose synthesis could be explored in the future. With new emerging characterization technologies, controlled and comparative synthesis of algal-based NPs could be done, which will help in refining the qualities of algal-mediated NPs for their industrial applications.

## Figures and Tables

**Figure 1 biomolecules-10-01498-f001:**
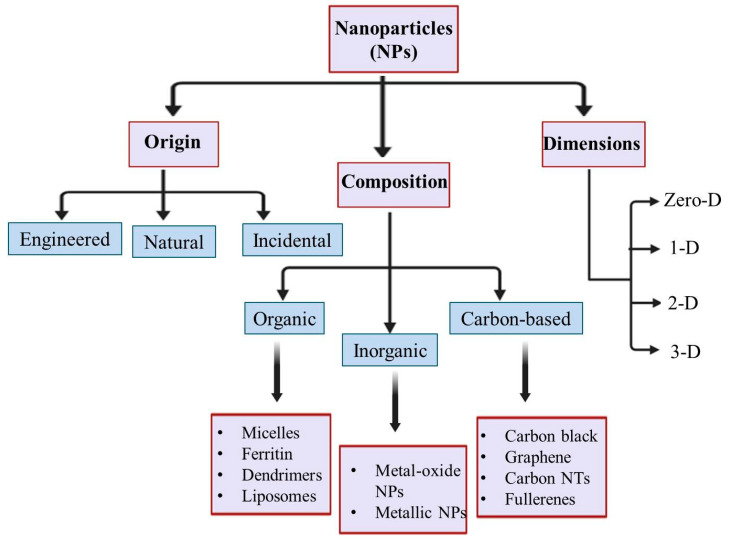
NPs (nanoparticles; usually ranging in size from 1 to 100 nm) classification on the basis of origin, dimensions, and structural configurations.

**Figure 2 biomolecules-10-01498-f002:**
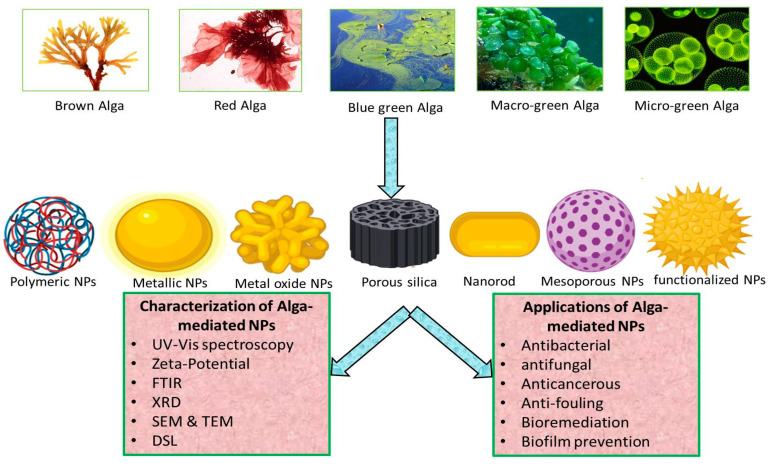
Graphical representation of algae-mediated biosynthesis, characterization and applications of NPs.

**Figure 3 biomolecules-10-01498-f003:**
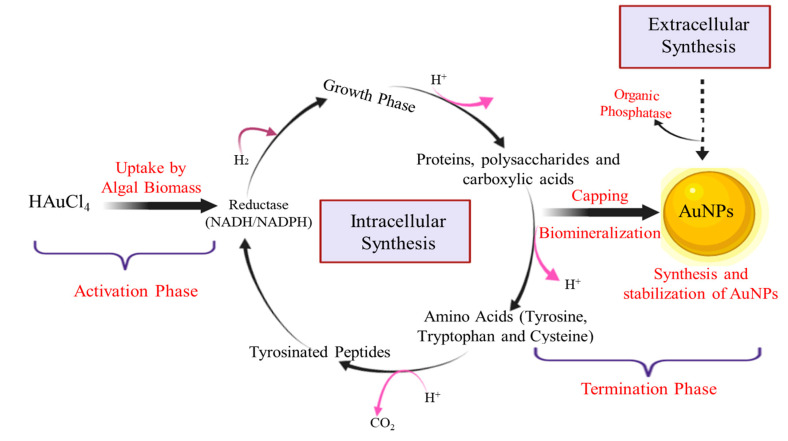
Schematic illustration of mechanism involved in extracellular and intracellular synthesis of algae-mediated gold nanoparticles (AuNPs).

**Figure 4 biomolecules-10-01498-f004:**
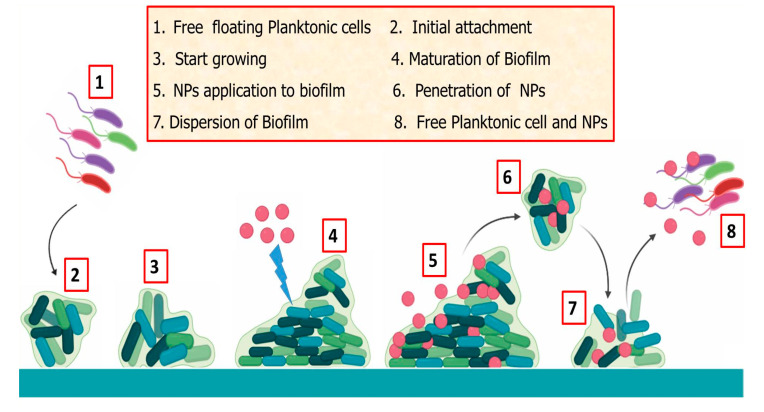
Graphical representation of the antifouling activity of the algae-mediated NPs.

**Table 1 biomolecules-10-01498-t001:** Brown algae-mediated biosynthesis of metallic NPs.

Algae	NPs	Location	Synthesis Conditions	Shape and Size	Characterization	Reducing Agent	References
*Turbinaria conoides*	Ag	Extracellular	1 h incubation in dark	Spherical, 96 nm	SEM, TEM, FTIR, and XRD	Carbonyl groups and polyamines	[44]
*Sargassum* spp.	Au	Extracellular	Neutral pH, 5 h incubation at room temperature	Longest edges, hexagonal and triangular, 300–400 nm	AFM, XRD, FTIR, and UV-Vis	Hydroxyl, amine functional groups	[45]
*Fucus vesiculosus*	Au	Extracellular	pH range (2–9), incubation time (1–8 h)	Spherical, 20–50 nm	XRD, TEM, SEM, EDS, and FTIR	Hydroxyl groups present in polysaccharides.	[46]
*Sargassum polycystum*	Ag	Extracellular	-	Spherical, 7 nm	UV-Vis, XRD, FTIR, SEM, and TEM	Extracellular terpenoids	[47]
*Turbinaria ornata*	Ag	Extracellular	24 h incubation	Spherical, polydispersed, 22 nm	UV-Vis, EDS, FE-SEM, XRD, and FTIR	Organic moieties	[48]
*Sargassum muticum*	Ag	Extracellular	2 h incubation at room temperature	Spherical, crystalline, 43–79 nm	UV–Vis, SEM, FTIR, EDX, and XRD	Sulfate and hydroxyl groups	[49]
*Sargassum muticum*	ZnO	Extracellular	Heating at 450 °C for 4 h	Hexagonal, 30–57 nm	UV–Vis, SEM, FESEM, FTIR, and XRD	Sulfate and hydroxyl groups	[11]
*Laminaria Japonica*	Au	Extracellular	-	Spherical, 15–20 nm	UV-Vis, XRD, FTIR, and TEM	Polysaccharides	[50]
*Turbinaria conoides*	Au	Extracellular	24 h incubation	Polydispersed, rectangular, triangular, 6–10 nm	UV-Vis, FTIR, and TEM	Carboxylic, fucoidan and polyphenolic groups	[51]
*Ecklonia cava*	Au	Extracellular	1 min incubation at 80 °C on magnetic stirrer	Spherical, 20–50 nm	UV-Vis, FESEM, TEM, XRD, and FTIR	Hydroxyl and phenolic groups	[52]
*Gilidiella acerosa*	Ag	Extracellular	-	Spherical 18–46 nm	UV-Vis, XRD, HR-TEM, and SEM	Metabolites	[53]
*Sargassum wightii*	Au	Extracellular	12–15 h continuous stirring	Thin planner, 8–12 nm	UV-Vis, XRD and HR-TEM	-	[54]
*Sargassum muticum*	Ag Au	Extracellular	-	Spherical, 41.0 ± 5.7 nm 10.4 ± 1.2 nm	UV-Vis, XRD, and HR-TEM	Polysaccharides	[55]
*Sargassum myriocystum*	Au	Extracellular	15 min incubation in water bath (72 °C)	Polydispersed, spherical and triangular, 15 nm	GC-MS, EDAX, XRD, UV-Vis, and FTIR	1-cyclopentyl group	[56]
*Turbinaria conoides*	Au	Extracellular	1 h of continuous stirring at room temperature	Rectangular, square, and triangular, 15 nm	EDS and SEM	Biochemical groups	[44]
*Stoechospermum marginatum*	Au	Extracellular	2 h incubation	Hexagonal, triangular and spherical, 18.7–93.7 nm	TEM, XRD, WDXRF, SEM, and FTIR	Hydroxyl groups	[57]
*Padina gymnospora*	Au	Extracellular	Temperature range of 24–95 °C, incubation of (0–20 h) and pH (4–12)	Spherical, 8–21 nm	AFM, UV-Vis, FTIR, TEM, and XRD	Fucoxanthin and flavonoids	[58]
*Dictyota bartayresiana*	Au	Extracellular	-	Polydispersed and spherical nanoparticles	FTIR, SEM, and UV-Vis	Phenolics, carboxylic and amine groups	[59]
*Ecklonia cava*	Au	Extracellular	1 min at 80 °C	Spherical, FCC, and triangular, 20–50 nm	FTIR, FESEM, EDX, and UV-Vis	Amine and hydroxyl groups	[52]
*Padina gymnospora*	Au	Extracellular	Temperature range of 24–95 °C, incubation of (0–20 h) and pH (4–12)	Spherical, 8–21 nm	AFM, UV-Vis, XRD, and HR-TEM	Fucoxanthin and flavonoids	[58]
*Sargassum muticum*	Au	Extracellular	1 h stirring at 45 °C	FCC and spherical, 5.42 nm	Zeta potential, UV-Vis, and TEM	Proteins and bio-organic compounds	[39]
*Sargassum tenerrimum*	Au	Extracellular	90 min incubation	Anisotropic and poly-dispersed, 4–45 nm	HR-TEM, UV-Vis, FTIR, DLS, and zeta potential	Secondary metabolites	[60]
*Cystoseira baccata*	Au	Extracellular	24 h stirring at room temperature	Poly-crystalline and spherical, 8.4 ± 2.2 nm	Zeta potential, UV-Vis, EELS, EDS, FTIR, and TEM	Poly-phenols and polysaccharides	[61]
*Sargassum wightiigrevilli*	Ag	Extracellular	120 h incubation at 37 °C, 5.6 pH	Spherical, 8–27 nm	UV-Vis, HR-TEM, XRD, and FTIR	Oxidation of alcoholic group	[62]
*Sargassum ilicifolium*	Ag	Extracellular	-	Spherical, 33–40 nm	UV-Vis, SEM, and TEM	Bio-active compounds	[63]
*Sargassum polycystum*	Ag	Extracellular	24 h incubation at room temperature	FCC and spherical, 5–7 nm	UV-Vis, FTIR, GC-MS, and HR-TEM	Hexadecane and octadecanol	[64]
*Sargassum plagiophyllum*	Ag	Intracellular	-	Spherical, 20–50 nm	SEM, FTIR, and UV-Vis	Carboxylic and benzene rings	[65]
*Padina pavonica*	Ag	Extracellular	24 h incubation at room temperature	Polydispersed and spherical, 10–72 nm	UV-Vis, XRD, SEM, and TEM	Terpenoids	[41]
*Padina tetrastromatica1*	Ag	Extracellular	48 h incubation on magnetic stirrer at 60 °C	Spherical, 4 nm	FTIR, TEM, and UV-Vis	Alkanes	[66]
*Padina gymnospora*	Ag	Extracellular	-	Spherical, 25–40 nm	UV-Vis and TEM	Aqueous extract	[67]
*Colpmenia sinusa*	Ag	Extracellular	Continuous stirring for 20 min at 70 °C, pH 10	Spherical, 20 nm	TEM, FTIR, GLC-C-O, and UV-Vis	Polysaccharides	[68]
*Cystophora moniliformis*	Ag	Extracellular	30 min incubation at 65 °C	FCC, 75 nm	SEM, DLS, EDX, and zeta-potential	Metabolites, phenolic compounds	[42]
*Sargassum cinereum*	Ag	Extracellular	3 h incubation and centrifuged at 13,000 rpm	Triangular and spherical 45–76 nm	UV-Vis, and SEM	-	[69]
*Sargassum longifolium*	Ag	Extracellular	-	Cubical, 30 nm	UV-Vis, SEM, FTIR, and EDS	Ketone and carboxylic groups	[70]
*Scaberia agardhii*	Ag	Extracellular	24 h incubation	Polydispersed, 40–50 nm	UV-Vis, SEM, EDAX, and FTIR	Proteins	[71]
*Sargassum polycystum C. Agardh*	Ag	Extracellular	-	Spherical	UV-Vis, XRD, and FTIR	Metabolites and proteins	[72]
*Sargassum vulgare*	Ag	Extracellular	-	Spherical, 10 nm	TEM, XRD, HR-TEM, EDX, and FACS	Secondary OH groups	[73]
*Turbinaria ornate*	Ag	Extracellular	Incubation at 70 °C water bath and centrifugation at 10,000 rpm	Polydispersed and spherical, 22 nm	UV-Vis, SEM, FESEM, and XRD	Organic moieties	[74]
*Sargassum muticum*	Ag	Extracellular	90 min incubation	Crystalline and spherical, 43–79 nm	UV-Vis, FTIR, SEM, EDX, and XRD	Sulfate and hydroxyl groups	[75]

**Table 2 biomolecules-10-01498-t002:** Red algae-mediated biosynthesis of metallic NPs.

Algae	NPs	Location	Synthesis Conditions	Shape and Size	Characterization	Reducing Agent	References
*Gracilaria edulis*	Ag	Extracellular	Incubation on orbital shaker at 150 rpm under 40 °C	Spherical, 12.5–100 nm,	UV-Vis, SEM, TEM, XRD, FTIR	Proteins and terpenoids	[88]
*Gelidium amansii*	Ag	Extracellular	48 h incubation at room temperature and centrifugation at 13,000 rpm	Spherical	UV-Vis	-	[89]
*Kappaphycus alvarezii*	Au	Extracellular	-	Polydispersed, 10–40 nm	XRD, TEM, FTIR FAAS, and UV-Vis	Polyphenolic compounds	[90]
*Galaxaura elongate*	Au	Extracellular	Incubation (10 min–12 h) at continuous stirring at 120 rpm	Rod, truncated and triangular shaped, 3.85–77.13 nm	Zeta-potential, TEM, HPLC, and GC-MS	Palmitic acid	[91]
*Chondrus crispus*	Au	Extracellular	Room temperature stirring, pH 2, 4, and 10	Spherical and polyhedral, 30–50 nm	UV-Vis, SEM, EDS, TEM, FAAS, and FTIR	Proteins	[92]
*Lemanea fluviatilis*	Au	Extracellular	12 h room temperature stirring	Polydispersed, FCC and spherical, 5–15 nm	UV-Vis, XRD, FTIR, DLS, and TEM	Protein and organic molecules	[93]
*Gracilaria corticata*	Au	Extracellular	Room temperature stirring	Spherical, 45–57 nm	UV-Vis and SEM		[94]
*Corallina officinalis*	Au	Extracellular	-			Functional groups	[95]
*Gelidiella acerosa*	Ag	Extracellular	48 h incubation on 120 rpm continuous stirring	FCC and spherical, 22 nm	UV-Vis, SEM, XRD, FTIR, and TEM	Aromatic compounds	[96]
*Acanthophora spicifera*	Ag	Extracellular	20 min of incubation at 60 °C	Spherical, 48 nm	TEM, FTIR, and UV-Vis	Alcoholic, carboxylic acid and phenolic compounds	[97]
*Gracilaria dura*	Ag	Extracellular	Incubation period (1, 4, and 48 h), temperature (25, 60, and 100 °C) and pH 6	Spherical, 6 nm	EDX, XRD, SAED, TGA, TEM, and TGA	Polymers	[98]
*Gelidiella* sp.	Ag	Extracellular	10 min incubation at 121 °C and centrifugation at 10,000 rpm	Spherical, 50 nm	UV-Vis, XRD, EDS, SEM, and FTIR	Proteins	[99]
*Kappa phycus* sp.	Ag	Extracellular	-	52–104 nm	UV-Vis, AFM, and FTIR		[100]
*Gracilaria corticata*	Ag	Extracellular	20 min incubation at 60 °C heating mantle	46 nm	UV-Vis, TEM, FTIR, DLS, and zeta-potential	Phenolic and amides	[101]
*Kappaphycus alverazii*	Ag	Extracellular	96 h of incubation at 27 °C at 250 rpm in orbital shaker	FCC, 73 nm	UV-Vis, SEM, FTIR, and EDX	Polysaccharides and functional groups	[102]
*Pterocladia capillacae*	Ag	Extracellular	20 min magnetic stirring at 70 °C	Spherical, 7 nm	UV-Vis, TEM, GLC, and FTIR	Carboxylic and sulfate groups	[68]
*Gracilaria birdiae*	Ag	Extracellular	30 min incubation at 0 °C at pH 10	Spherical, 20.3 nm	DLS, FTIR, TEM, UV-Vis, and zeta-potential	Hydroxyl and carbonyl groups	[103]
*Jania rubins*	Ag	Extracellular	20 min magnetic stirring at 70 °C	Spherical, 12 nm	TEM, FTIR, GLC, and UV-Vis	Carbonyl groups	[68]
*Acanthophora specifera*	Ag	Extracellular	Room temperature incubation	Cubic, 81 nm	FTIR and XRD	Polysaccharides and uronic acids	[104]
*Amphiroa fragilissima*	Ag	Extracellular	20 min incubation, centrifugation at 12,000 rpm	Crystalline	UV-Vis, XRD, and FTIR	Peptide	[105]
*Porphyra vietnamensis*	Ag	Extracellular	15 min incubation at 70 °C, pH 11	13 ± 3 nm	FTIR	Polysaccharides	[87]
*Desmarestia menziesii*	Ag Au	Extracellular	-	7.0 ± 1.2 nm 17.8 ± 2.6 nm	FTIR, UV-Vis, SEM, and TEM	Carbonyl groups	[106]
*Gracilaria* sp.	Ag-Au	Extracellular	Overnight incubation at 60 °C, pH 6	Crystalline	FTIR, UV-Vis, SEM, and TEM	Functional groups	[107]

**Table 3 biomolecules-10-01498-t003:** Blue-green algae-mediated biosynthesis of NPs.

Algae	NPs	Location	Shape and Size (nm)	Synthesis Conditions	Characterization	Reducing Agent	References
*Nostoc ellipsosporum*	Au	Extracellular	Decahedral and icosahedron, 20–40 nm	3 h incubation, pH 5	UV-Vis, SEM, and FT-IR	Proteins and carboxylate groups	[116]
*Spirulina platenesis*	Au	Extracellular	Monodispersed and spherical, 2–8 nm	-	FT-IR, UV-Vis, HR-TEM, and EDAX	Proteins	[117]
*Spirulina platensis*	Au	Extracellular	Spherical, 20–30 nm	40 h incubation	UV-Vis, FTIR, TEM, SEM-EDAX, XRD, NAA, AAS	Biomolecules (amino, carboxylic, phosphate, thiol)	[40]
*Spirulina platensis*	Au	Extracellular	Octahedral and cubic	48 h incubation at room temperature, centrifugation at 10,000 rpm	UV-Vis and SEM	Proteins and peptides	[118]
*Plectonema boryanum*	Au	Extracellular	20–25 nm	24 h incubation at 200 °C	TEM, XPS, TOF-SIMS, and SEM	-	[119]
*Synechocystis* sp.	Au	Extracellular	Spherical, 3–13 nm	16 h incubation at 20 °C in light (50 μmol photons m^−2^ s^−1^), dark and 4 °C in dark respectively	TEM, SERS, and zeta-potential	Carboxylic groups and polysaccharides	[120]
*Lyngbya majuscula* and *Spirulina subsalsa*	Au	Intracellular	Spherical, 20 nm	72 h incubation, pH 6, 7 and 8	TEM	-	[121]
*Phormidium tenue*	Au	Extracellular	Spherical and irregular, 14.84 nm	72 h exposure, pH 7 and 0	TEM, UV-Vis, and XRD	Enzymes and metabolites	[122]
*Phormidium* sp.	Au	Intracellular	Triangular, 25 nm	Incubation at room temperature	UV-Vis, HR-SEM, TEM, FT-IR, and EDX	Cytoplasmic proteins	[123]
*Phormidium* sp.	Au	Intracellular	Monodispersed, Triangular	-	UV-Vis, SEM, and TEM	Proteins	[124]
*Phormidium valderianum*	Au	Extracellular	Spherical, hexagonal, and FCC, 24 nm	72 h incubation at 20 °C	TEM, UV-Vis, and XRD	Metabolites	[122]
*Lyngbya majuscule*	Au	Extracellular	Hexagonal and spherical, 2–25 nm	Incubation at room temperature and centrifugation 10,000 rpm	UV-Vis, MLDLS, FT-IR and TEM	Proteins	[125]
*Anabaena* sp.	Au	Extracellular	Spherical, 10 nm	4–40 h incubation	TEM and XRD	Protein moieties	[126]
*Spirulina subsalsa*	Au	Extracellular	Spherical, 30 nm	-	UV-Vis, XRD, FT-IR, and MLDLS	Extracellular proteins	[125]
*Oscillato riawillei*	Ag	Extracellular	Spherical, 10–25 nm	24 h incubation	UV-Vis, EDS, FT-IR, and SEM	Tryptophan	[127]
*Spirulina platensis*	Ag	Extracellular	Spherical, 2–8 nm	48 h incubation	SEM and UV-Vis	Phytochemicals	[128]
*Plectonemaboryanum*	Ag	Intracellular	Octahedral, 200 nm	25–100 °C incubation upto 28 days	XPS, EDS, and TEM	-	[129]
*Aphanothece* sp. and *Oscillitoria* sp.	Ag	Extracellular	Spherical, 44–79 nm	-	UV-Vis, EDX, and SEM	Bio-active compounds	[130]
*Microchaete*	Ag	Extracellular	Polydispersed and spherical, 80 nm	60 min incubation at 60 °C, pH 5.6	TEM	Cellular metabolites	[131]
*Cylindrospermum stagnale*	Ag	Extracellular	Pentagonal, 38–88 nm	45 h incubation at 40 °C	UV-Vis and SEM	Extracellular proteins	[132]
*Spirulina platensis*	Si	Extracellular	Crystalline	24 h of incubation at 25 °C, pH 7	UV-Vis	-	[133]
*Spirulina platensis*	Au-Ag	Extracellular	Core-shell	120 h of incubation 37 °C, pH 5.6	UV-Vis and SEM	Bio-active compounds	[62]
*Chlamydomonas reinhardtii*	CdSNPs	Extracellular	-	Incubation at 65 °C, centrifugation	UV-Vis and SEM	Extracellular proteins	[134]

**Table 4 biomolecules-10-01498-t004:** Micro green algae-mediated biosynthesis of Metallic NPs.

Algae	NPs	Location	Shape and Size	Synthesis Conditions	Characterization	Reducing Agent	References
*Chlorella vulgaris*	Au	Extracellular	9–20 nm	3 h incubation for 45–90 °C	XRD, FTIR, SEM, UV-Vis	Biomolecules (amino, carboxylic, phosphate, thiol)	[142]
*Scencedesmus* sp.	Ag	Extracellular	15–20 nm	Incubation at 37 °C on 200 rpm shaking	XRD, FTIR, SEM, UV-Vis	-	[143]
*Pithophora oedogonia*	Ag	Extracellular	Cubical and hexagonal, 24–55 nm	Incubation at room temperature, centrifugation at 15,000 rpm for 15 min	UV-Vis, SEM, EDS, and DLS	Proteins and functional groups	[13]
*Phormidiumtenue*	Au	Extracellular	Spherical- and irregular-shaped, 14.84 nm,	-	UV-Vis, TEM, XRD	Cellular metabolites	[144]
*Plectonema boryanum*	Ag	Intracellular	Less than 10 nm	Incubation at 25 °C	TEM, XPS, TEM-ED	Nitrate ions	[47]
*Chlorococcum humicola*	Ag	Intracellular	Spherical, 16 nm	72 h incubation at 28 °C, centrifugation at 12,000 rpm	UV-Vis, XRD, SEM, EDX, TEM, and FTIR	Intracellular protein molecules	[145]
*Enteromorpha flexuosa*	Ag	Extracellular	Spherical, 2–32 nm	1 h incubation	UV-Vis, XRD, SEM, EDX, EDS and FTIR	Functional groups	[141]
*Chlorella vulgaris*	Au	Extracellular	Triangular and truncated	48 h of incubation	FESEM, FTIR, UV-Vis, SAED, and AFM	Proteins	[146]
*Klebsormidium a flaccidum*	Au	Intracellular	10–20 nm	Total 100 h incubation	UV-Vis, PAM, SERS, XPS, and TEM	In thylakoids	[33]
*Chlorella pyrenoidusa*	Au	Extracellular	Icosahedral and spherical, 25–30 nm	Incubation at 100 °C at 100 rpm, pH 8	XRD, HR-TEM, and UV-Vis	Enzymes	[37]
*Tetraselmis suecica*	Au	Extracellular	Spherical, 120 nm	5 min incubation at 90 °C	XRD, FTIR, and UV-Vis	Hydroxyl, carbonyl, and nitrate function groups	[24]
*Tetraselmis kochinensis*	Au	Intracellular	Triangular, FCC, and spherical, 5–35 nm	48 h incubation at 28–29 °C on 200 rpm shaker	UV-Vis and TEM	Cell wall and cytoplasm’s enzymes	[34]
*Coelastrella* sp.	Au	Intracellular	Spherical, 30 nm	1 h incubation in dark, centrifugation at 6,000 rpm for 7 min	UV-Vis, EDAX, SEM, TEM, and FTIR	Cytoplasm’s micro-proteins	[147]
*Cosmarium impressulum*	Au	Extracellular	Spherical, 5.7 nm	Incubation of 16 h light and 8 h dark at 20 °C	TEM and UV-Vis	Enzymes	[16]
*Spirogyra submaxima*	Au	Intracellular	Triangular and spherical	-	UV-Vis, TEM, XRD and zeta potential	At cytoplasm and chloroplast	[148]
*Chlorella vulgaris*	Ag	Extracellular	Triangular, 28 nm	12 h incubation at room temperature	UV-Vis, HR- TEM and FESEM	Carbonyl groups	[149]
*Nannochloropsis oculata* and *Chlorella vulgaris*	Ag	Extracellular	FCC, 15 nm	24–48 h of incubation	XRD and TEM	Proteins	[150]
*Chlamydomonas reinhardtii*	Ag	Extracellular	Rectangular and rounded, 1–15 nm	5–10 h of incubation	SEM, EDAX, MALDI-TOF-MS and UV-Vis	Cellular proteins like histones	[151]
*Chlorococcum humicola*	Ag	Intracellular	Spherical, 16 nm	48 h of incubation at 24 °C	XRD, EDX, SEM, FTIR, and UV-Vis	Intracellular proteins	[152]
*Euglena gracilis*	Ag	Extracellular	Polydispersed and spherical, 15–60 nm	1–4 h incubation, pH 7.5	UV-Vis, ICP-AES, EDAX, SEM, FTIR, and TEM	Primary proteins	[153]
*Chlorella pyrenoidosa*	Ag	Extracellular	FCC, 5–20 nm	24 h incubation at 28 °C, centrifugation at 10,000 rpm for 15 min	XRD, SEM, TEM, EDX, UV-Vis, and FTIR	Cyclic peptides	[154]
*Euglena intermedia*	Ag	Extracellular	Polydispersed and spherical, 24 nm	1–4 h incubation, pH 7.5	UV-Vis, ICP-AES, EDAX, SEM, FTIR, and TEM	Proteins	[153]
*Enteromorpha flexuosa*	Ag	Extracellular	Circular, 15 nm	1 h incubation	UV-Vis, TEM, EDS, XRD, and FTIR	Amines and peptides	[141]
*Ulva lactuca*	Ag	Extracellular	Spherical, 20–50 nm	-	UV-Vis, TEM, XRD, TGA, EDAX, and SEM	Protein molecules	[155]
*Caulerpa racemosa*	Ag	Extracellular	FCC, 5–25 nm	24 h incubation at room temperature	XRD, TEM, FTIR, and SEM	Cyclic peptides	[156]
*Prasiola crispa*	Au	Extracellular	Spherical, 5–25 nm	-	UV-Vis, XRD, FTIR, DLS, HR-TEM	Proteins	[125]
*Pithophora oedogonia*	Ag	Extracellular	Cubical and hexagonal, 25–44 nm	Incubation at room temperature, centrifugation at 15,000 rpm for 15 min	EDS, UV-Vis, DLS, and FTIR	Phytochemicals and proteins	[13]
*Caulerpa serrulata*	Ag	Extracellular	Spherical and FCC, 10 nm	Different time and pH ranges were studied	HR-TEM, FTIR, XRD, and UV-Vis	-	[19]
*Chaetomorpha linum*	Ag	Extracellular	Coalescence, 3–44 nm	30 min of incubation	UV-Vis, FTIR, SEM, EDX, and TEM	-	[157]
*Codium captitatum*	Ag	Extracellular	Coalescence, 3–44 nm	48 h room temperature incubation	UV-Vis, DLS, FTIR, and TEM	Carboxylic group	[158]
*Acutodesmus dimorphus*	Ag	Extracellular	Spherical, 2–20 nm	24 h incubation, centrifugation at 16,000 rpm for 20 min at 4 °C	UV-Vis, FTIR and TEM	Proteins	[159]
*Botryococcus braunii*	Ag	Extracellular	Spherical, 16 nm	3 h continuous stirring at room temperature, dried AgNPs at 55 °C for 5 h	-	Peptides and amines	[160]
*Cholera vulgaris*	Si	Extracellular	Spherical	13 days slow incubation	UV-Vis, TEM, and SEM	Peptides and proteins	[161]
*Prasiola crispa*	Au	Extracellular	FCC, facile, and spherical, 25 nm	12 h incubation, centrifugation at 10,000 rpm	UV-Vis, XRD, FTIR, and DLS	Proteins	[162]

**Table 5 biomolecules-10-01498-t005:** Macro green algae-mediated biosynthesis of Metallic NPs.

Algae	NPs	Location	Shape and Size	Synthesis Conditions	Characterization	Reducing Agent	References
*Rhizoclonium fontinale*	Au	Intracellular	Spherical (5–20 nm), nano-triangles (15–88 nm)	72 h incubation, pH 9	UV-Vis, HR-TEM, DLS, EDAX	Intracellular synthesis	[65]
*Rhizoclonium fontinale*	Au	Intracellular	Spherical (5–20 nm) and hexagonal (34 nm)	Incubation intervals (1, 3, 24, 48 and 72 h) pH 9	UV-Vis, HR-TEM, EDAX, DLS, and SEM	Intracellular proteins	[18]
*Ulva reticulata*	Ag	Extracellular	Spherical, 40–50 nm	Incubation at room temperature	UV-Vis, FT-IR, SEM, XRD	Carboxylic acids, benzene rings, and fluoro alkane	[158]
*Ulva intestinalis*	Ag Au	Extracellular	-	Incubation at room temperature	UV-Vis, HR-TEM, EDAX, and SEM	Polysaccharides	[165]
*Enteromorpha compressa4*	Ag	Extracellular	Spherical, 0–50 nm	-	UV-Vis, FT-IR, SEM, XRD	Benzene rings and hydrogen bonded alcohols	[158]
*Codium capitatum*	Ag	Extracellular	3–44 nm	Incubation in dark at room temperature	UV-Vis, EDX, FT-IR	Amine, peptide, and sulfate groups	[163]
*Gracilaria edulis*	Ag and ZnO	Extracellular	55–99 nm and 66–95 nm	Centrifugation at 10,000 rpm	UV-Vis, XRD, EDX, FT-IR, and FESEM	Alcohol, amides, and nitro groups	[163]
*Urospora* sp.	Ag	Extracellular	FCC and spherical, 20–30 nm	Incubation in dark at 70 °C on magnetic stirrer	FT-IR, HR-TEM, XRD, and UV-Vis	Hydroxyl and carbonyl groups	[166]
*Chaetomorpha linum*	Ag	Extracellular	3–44 nm	Incubation at 37 °C at static condition	SEM, FT-IR, and UV-Vis	Peptides, flavonoids and terpenoids	[157]
*Ulva faciata*	Ag	Extracellular	Spherical, 7–20 nm	20 mi of continuous stirring at 70 °C	TEM, UV-Vis, FT-IR, and GLC.	Polyphenolic groups	[68]
*Rhizoclonium fontinale*	Au	Intracellular	Spherical, 5–20 nm and hexagonal, 34 nm	72 h of incubation, pH 9	UV-Vis, HR-TEM, EDAX, DLS, and SEM	Intracellular proteins	[18]
*Ulva reticula*	Ag	Extracellular	Spherical, 40–50 nm	-	HR-TEM, FT-IR, XRD, and UV-Vis	-	[65]
*Ulva faciata*	Ag	Extracellular	-	-	UV-Vis	Proteins	[68]
*Ulva flexousa*	Ag	Extracellular	Circular and FCC, 2–32 nm	24 h incubation in illuminated conditions, centrifugation at 5000 rpm	XRD, TEM, FT-IR, and UV-Vis	Peptides	[167]
*Spirogyra varians*	Ag	Extracellular	FCC, 17.6 nm	20 min incubation in dark	UV-Vis, SEM, and FT-IR	Amino, carboxylic, and hydroxyl groups	[164]
*Spirogyra insignis*	Ag	Extracellular	Spherical, 28–41 nm	pH 2, 4 and 10	UV-Vis, FT-IR, and TEM	Proteins	[92]

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
