# Peer review of "An Overview of the Algae-Mediated Biosynthesis of Nanoparticles and Their Biomedical Applications"

_biomolecules, 2020, doi:10.3390/biom10111498_

Round 1
Reviewer 1 Report
The authors collected a lot of literature (187 literary sources) in the article. I appreciate this work because it requires a lot of time and effort. The manuscript is, in fact, just an overview of experiments already done on the preparation of silver nanoparticles by means of various types of algae.
In the article, I miss some specific benefits, for example for science or recommendations for further experiments. The fact that algae are able to synthesize silver nanoparticles is well known (to scientists working in the field). Even though it is an overview, the authors should process the data so that the article also has some scientific benefit.
There are also some inaccuracies in the article:
Introduction - is too general, there are many well-known definitions. For example. definition of nanoparticles or how they are classified ...
Chapter 2.4 - it would be good to define exactly what is micro and what is macro size. Last line 223 of this chapter: "green algae as micro-mediated and micro-mediated". There are twice micro.
Figure 2 is of poor quality, difficult to read.
Chapter 4.1. line 330-331: ROS is reactive oxygen stress not species.
Despite the shortcomings, I recommend this article for publication.
Author Response
Reviewer-1
Comments: The authors collected a lot of literature (187 literary sources) in the article. I appreciate this work because it requires a lot of time and effort. The manuscript is, in fact, just an overview of experiments already done on the preparation of silver nanoparticles by means of various types of algae. In the article, I miss some specific benefits, for example for science or recommendations for further experiments. The fact that algae are able to synthesize silver nanoparticles is well known (to scientists working in the field). Even though it is an overview, the authors should process the data so that the article also has some scientific benefit.
AUTHORS: Thank you very much for your comments and suggestions that greatly help us to improve the quality of the present revised version of our MS. We do our best to answer to all your queries and hope this revised version will answer them. Revisions appear in track changings in the revised manuscript.
Sir, we have critically analyzed the published data and identify the knowledge gaps existing in this field and incorporated in the MS. Moreover, we have also incorporated the data about limitations of algal-biosynthesis and recommendations for future experiments in section: 5 (limitations and future prospects).
- There are also some inaccuracies in the article: Introduction - is too general, there are many well-known definitions. For example. definition of nanoparticles or how they are classified ...
AUTHORS: We have revised the classification of nanoparticles in the introduction and describe it diagrammatically in figure 1.
- Chapter 2.4 - it would be good to define exactly what is micro and what is macro size. Last line 223 of this chapter: "green algae as micro-mediated and micro-mediated". There are twice micro.
AUTHORS: Sir, we have defined the micro and macro algae in respective section at line 311-312. Secondly, we have corrected the sentence as micro and macro-mediated at line -315.
- Figure 2 is of poor quality, difficult to read.
AUTHORS: We have redrawn the figure to improve its quality.
- Chapter 4.1. Line 330-331: ROS is reactive oxygen stress not species.
AUTHORS: Respected sir, we added ROS as reactive oxygen species and in all online resources ROS is abbreviated as reactive oxygen species not the reactive oxygen stress.
- Despite the shortcomings, I recommend this article for publication.
AUTHORS: Thank you very much Sir and we have tried our best to improve the quality of the manuscript.
Reviewer 2 Report
This review focuses in an upcoming hot topic that does not have many recent reviews of the innumerous developments. Overall this manuscript presents an extensive compilation that will be of interest to the community.
There are some typos and aqward sentences that should be reviewed throughout the manuscript (e.g. “While, one dimensional NPs possess…”, “Zero dimensional NPs have their breath…”).
The authors describe the different possibilities of nanoparticle classification in a continuous, poorly structured way, which makes it confusing as an introduction to the topic. A restructuring or inclusion of an illustrative scheme would improve this.
Table 1 legend should clearly state it lists only metallic nanoparticles.
Despite the intense search of bibliography, some of the most recent references may be missing (e.g. https://doi.org/10.1039/C9TB00215D, https://doi.org/10.1007/s40097-020-00352-y, https://doi.org/10.1016/j.polar.2017.10.004). Please update.
The structure should present the mechanisms of nanoparticle formation through this methodology before listing the existing published reports. For example, Figure 2 would be more helpful before tables 4-5.
The description of the state-of-the-art of nanoparticles with anti-cancerous activity is too simplistic and does not translate the intense development of this approach.
Author Response
Reviewer-2
Comments: This review focuses in an upcoming hot topic that does not have many recent reviews of the innumerous developments. Overall this manuscript presents an extensive compilation that will be of interest to the community.
AUTHORS: Thank you very much for your comments and suggestions that greatly help us to improve the quality of the present revised version of our MS. We do our best to answer to all your queries and hope this revised version will answer them. Revisions appear in track changings in the revised manuscript.
- There are some typos and aqward sentences that should be reviewed throughout the manuscript (e.g. “While, one dimensional NPs possess…”, “Zero dimensional NPs have their breath…”).
AUTHORS: Sir we have removed this sentence from the revised MS and according to your suggestions an illustrative scheme regarding the classification of NPs has been added as Figure 1 in the manuscript.
- The authors describe the different possibilities of nanoparticle classification in a continuous, poorly structured way, which makes it confusing as an introduction to the topic. A restructuring or inclusion of an illustrative scheme would improve this.
AUTHORS: Thank you for this valuable comment. We correct it accordingly and added illustrative scheme as figure 1 in introduction section.
- Table 1 legend should clearly state it lists only metallic nanoparticles.
AUTHORS: Sir we have revised the legend of table 1 according to your suggestion.
- Despite the intense search of bibliography, some of the most recent references may be missing (e.g. https://doi.org/10.1039/C9TB00215D, https://doi.org/10.1007/s40097-020-00352-y, https://doi.org/10.1016/j.polar.2017.10.004). Please update.
AUTHORS: Sir we have added these studied as reference number 55, 106 and 165 in the revised manuscript. The data from these studies have also been added in corresponding tables as well. Moreover, beside these studies many other recent studies on this topic have been cited in the text especially in the biomedical applications section.
- The structure should present the mechanisms of nanoparticle formation through this methodology before listing the existing published reports. For example, Figure 2 would be more helpful before tables 1-5.
AUTHORS: Thank you for this valuable comment. We have added the mechanism of synthesis and now figure 3 before tables 1-5 in the revised MS.
- The description of the state-of-the-art of nanoparticles with anti-cancerous activity is too simplistic and does not translate the intense development of this approach.
AUTHORS: Sir, we have revised it accordingly and added all the current data published on anticancerous activities of algal-mediated nanoparticles. We try our best to improve this section with current studies and developments made in this field.
Reviewer 3 Report
1) The English language must be carefully revised.
2) Provide full names of techniques used to characterize nanoparticles. Briefly describe the possibilities of each technique.
3) In the tables add the conditions of biosynthesis (pH, temperature, time, etc.).
4) Add information about the possibilities of controlling size and shape of nanoparticles during biosynthesis.
5) “The application of NPs in humans should be free of any toxicity which is ensured by NPs synthesized from algae.” - What is the evidence that algae-mediated biosynthesis ensures non-toxicity of NPs for humans? Have such studies been conducted? Why the toxicity of the nanoparticles themselves would change depending on the method of synthesis?
6) The limitations of biosynthesis should be described more fully.
7) Fragments for improvement:
“chloroauric acid as a precursor salt” – acid as a salt?
“the reduction of metallic silver to silver ions”
“intracellular synthesis of AuNPs by Tetraselmis cochinensis cell wall was confirmed by UV-Visible spectroscopy showing the presence of NPs in concentrated form next to the cell wall instead of cytoplasmic synthesis” – The UV-Vis technique does not allow for localization of nanoparticles.
“13+3nm” – Should it be 13±3 nm?
“75µl AgNPs formed a highest zone of inhibition (21mm) against E. coli while at 50μL AgNPs formed smallest zone of inhibition (10mm) against S. typhi” - This is not clear to the reader.
Author Response
Reviewer-3
- The English language must be carefully revised.
AUTHORS: Thank you for this remark. We have revised the English language throughout the revised MS.
- Provide full names of techniques used to characterize nanoparticles. Briefly describe the possibilities of each technique.
AUTHORS: Sir we have provided the full names of all characterizations techniques and also briefly describe the possibilities of each technique in the introduction section in lines 76-87.
- In the tables add the conditions of biosynthesis (pH, temperature, time, etc.).
AUTHORS: Sir we have revised all the tables according to your suggestions and new column has been added in each table describing the synthesis conditions.
- Add information about the possibilities of controlling size and shape of nanoparticles during biosynthesis.
AUTHORS: Sir we have briefly added the data about different factors that can control the size and shape of nanoparticles in the introduction section in lines 72-79.
- “The application of NPs in humans should be free of any toxicity which is ensured by NPs synthesized from algae.” - What is the evidence that algae-mediated biosynthesis ensures non-toxicity of NPs for humans? Have such studies been conducted? Why the toxicity of the nanoparticles themselves would change depending on the method of synthesis?
AUTHORS: Sir we have revised this section. Actually the green synthesized NPs have been capped by naturally biomolecules and did not carry any toxic chemicals that come from chemical synthesis. Therefore, the green synthesized (algal-mediated) nanoparticles are free from toxic chemicals entangled on their surfaces and thus most recommended for biomedical applications. Moreover, two recent studies have been carried out to evaluate the toxicity/biocompatibility of the algal-mediated nanoparticles. These studies showed that the algal-mediated nanoparticles are biocompatible. We have cited these studies in the manuscript and the links are given below;
- https://www.sciencedirect.com/science/article/abs/pii/S2213343718307905
- https://pubs.rsc.org/en/content/articlelanding/2019/TB/C9TB00215D#!divAbstract
- The limitations of biosynthesis should be described more fully.
AUTHORS: Sir, we have described the limitations of biosynthesis in section 5 (limitations and future prospects) in revised MS.
- “Chloroauric acid as a precursor salt” – acid as a salt?
AUTHORS: Thank you for pointing this mistake. We have corrected as metal precursor not the salt.
- “The reduction of metallic silver to silver ions”
AUTHORS: Thank you for pointing this mistake. We have corrected it as reduction of silver ions (Ag+) to zero valent silver (Ag0).
- “Intracellular synthesis of AuNPs by Tetraselmis cochinensis cell wall was confirmed by UV-Visible spectroscopy showing the presence of NPs in concentrated form next to the cell wall instead of cytoplasmic synthesis” Revise it.
AUTHORS: Sir, we have revised this sentence in the revised MS.
- “13+3nm” – Should it be 13±3 nm?
AUTHORS: Sir, we have corrected it in the corresponding table.
- “75µl AgNPs formed a highest zone of inhibition (21mm) against E. coli while at 50μL AgNPs formed smallest zone of inhibition (10mm) against S. typhi” - This is not clear to the reader.
AUTHORS: Sir, we have rephrased this sentence in the revised MS to make it clear to the readers.
Reviewer 4 Report
This manuscript consider for publication after major revision.
1)The author need to add minimum 10 more figure.
2)English correction is essential for this manuscript.
3) Author need to address Biomedical part with more current publications.
4) Mechanism description is very poor, image quality also.
5)Table description needs more perfect in text.
Author Response
Reviewer-4
- The authors need to add minimum 10 more figure.
AUTHORS: Thank you very much Sir for this remark. I am really sorry Sir, we cannot added 10 more figures to it, as it is a review and we have added one new figure as figure 1. Other than that we could not find any loophole where the figures can be added.
- English correction is essential for this manuscript.
AUTHORS: Sir we have revised the English language throughout the manuscript and try our best to make it more perfect.
- Author need to address biomedical part with more current publications.
AUTHORS: Sir we have revised the whole section of biomedical applications and updated each sub-section with current publications.
- Mechanism description is very poor; image quality also.
AUTHORS: Sir we have revised the mechanism description and also redraw the figure to make it more clear and improved its quality.
- Table description needs more perfect in text.
AUTHORS: Sir, We have revised the descriptions of tables according to your suggestion in the text.
Round 2
Reviewer 4 Report
Thanks. Your manuscript is accepted.
Author Response
Thank you very much for your useful comments.